# Genetic Variants in *KCTD1* Are Associated with Isolated Dental Anomalies

**DOI:** 10.3390/ijms25105179

**Published:** 2024-05-09

**Authors:** Cholaporn Ruangchan, Chumpol Ngamphiw, Annop Krasaesin, Narin Intarak, Sissades Tongsima, Massupa Kaewgahya, Katsushige Kawasaki, Phitsanu Mahawong, Kullaya Paripurana, Bussaneeya Sookawat, Peeranat Jatooratthawichot, Timothy C. Cox, Atsushi Ohazama, James R. Ketudat Cairns, Thantrira Porntaveetus, Piranit Kantaputra

**Affiliations:** 1Center of Excellence in Medical Genetics Research, Chiang Mai University, Chiang Mai 50200, Thailand; cholaporn_r@cmu.ac.th (C.R.); kwgmsp@gmail.com (M.K.); 2Division of Pediatric Dentistry, Department of Orthodontics and Pediatric Dentistry, Faculty of Dentistry, Chiang Mai University, Chiang Mai 50200, Thailand; 3National Biobank of Thailand, National Center for Genetic Engineering and Biotechnology (BIOTEC), Pathum Thani 12120, Thailand; chumpol.nga@gmail.com (C.N.); sissades.ton@nstda.or.th (S.T.); 4Center of Excellence in Genomics and Precision Dentistry, Department of Physiology, Faculty of Dentistry, Chulalongkorn University, Bangkok 10330, Thailand; annop.kss@hotmail.com (A.K.); narin.i@chula.ac.th (N.I.); 5Division of Oral Anatomy, Faculty of Dentistry & Graduate School of Medical and Dental Sciences, Niigata University, Niigata 950-2180, Japan; ka2shige@dent.niigata-u.ac.jp (K.K.); atsushiohazama@dent.niigata-u.ac.jp (A.O.); 6Division of Urology, Department of Surgery, Faculty of Medicine, Chiang Mai University, Chiang Mai 50200, Thailand; mahawongph@gmail.com; 7Dental Department, Suanphueng Hospital, Ratchaburi 70180, Thailand; jusuke07@hotmail.com (K.P.); peerajitsookawat@gmail.com (B.S.); 8School of Chemistry, Institute of Science, and Center for Biomolecular Structure, Function and Application, Suranaree University of Technology, Nakhon Ratchasima 30000, Thailand; pj.cbsfa@gmail.com (P.J.); cairns@sut.ac.th (J.R.K.C.); 9Departments of Oral & Craniofacial Sciences, School of Dentistry, and Pediatrics, School of Medicine, University of Missouri-Kansas City, Kansas City, MO 64110, USA; coxtc@umkc.edu

**Keywords:** root anomalies, hypodontia, tooth agenesis, taurodontism, supernumerary tooth, oral exostoses

## Abstract

KCTD1 plays crucial roles in regulating both the SHH and WNT/β-catenin signaling pathways, which are essential for tooth development. The objective of this study was to investigate if genetic variants in *KCTD1* might also be associated with isolated dental anomalies. We clinically and radiographically investigated 362 patients affected with isolated dental anomalies. Whole exome sequencing identified two unrelated families with rare (p.Arg241Gln) or novel (p.Pro243Ser) variants in *KCTD1*. The variants segregated with the dental anomalies in all nine patients from the two families. Clinical findings of the patients included taurodontism, unseparated roots, long roots, tooth agenesis, a supernumerary tooth, torus palatinus, and torus mandibularis. The role of Kctd1 in root development is supported by our immunohistochemical study showing high expression of Kctd1 in Hertwig epithelial root sheath. The KCTD1 variants in our patients are the first variants found to be located in the C-terminal domain, which might disrupt protein–protein interactions and/or SUMOylation and subsequently result in aberrant WNT-SHH-BMP signaling and isolated dental anomalies. Functional studies on the p.Arg241Gln variant are consistent with an impact on β-catenin levels and canonical WNT signaling. This is the first report of the association of *KCTD1* variants and isolated dental anomalies.

## 1. Introduction

During tooth development, tooth number, size, and position are regulated by the collaboration of WNT, SHH, FGF, and BMP signaling [1,2]. Normal root morphogenesis requires optimal levels of WNT/β-catenin and SHH signaling [2]. SHH is primarily found in the dental epithelium throughout tooth development, starting from initiation and continuing through root formation stages. It regulates the formation of enamel, dentin, cementum, and the surrounding periodontal structure [3]. In contrast, WNT/β-catenin signaling plays an essential role in neural crest cell survival, differentiation, and development of ectodermally derived structures, including skin, hair, mammary gland, nails, and teeth [4,5]. WNT signaling regulates various processes involved in odontogenesis, including tooth bud initiation, tooth crown morphogenesis, and tooth root development.

Several stages of tooth development require optimal WNT/β-catenin signaling [2]. Downregulation of WNT/β-catenin signaling in the dental epithelium is implicated in tooth agenesis or microdontia, whereas overactivation of WNT/β-catenin signaling in the dental epithelium is implicated in the formation of supernumerary teeth [1,6,7,8]. In contrast, upregulation of WNT signaling in the dental mesenchyme has been shown to result in tooth agenesis [9].

Genetic variants in the Potassium Channel Tetramerization Domain-Containing Protein 1 gene (*KCTD1*; MIM 613420) are implicated in Scalp-Ear-Nipple or Finlay–Marks syndrome (SEN; MIM 181270), which is characterized by cutis aplasia of the scalp, malformations of breasts, external ears, digits, and nails, as well as renal and dental anomalies [10,11,12,13,14]. Dental anomalies found in patients with Scalp-Ear-Nipple syndrome include microdontia, tooth agenesis, and enamel defects [10,13,15,16,17,18]. To date, thirteen genetic variants in *KCTD1*, all of which impact the N-terminus of the protein, have been reported to be associated with Scalp-Ear-Nipple syndrome [10,11,12,13,14].

The *KCTD1* gene encodes the KCTD1 protein, which directly binds to β-catenin through its BTB (Broad-complex, Tramtrack, and Bric-a-brac) domain, enhances its degradation, and inhibits canonical WNT/β-catenin signaling [19,20]. KCTD1 also functions as a transcriptional repressor, which interacts with transcription factor AP2-alpha (TFAP2A; MIM 107580), transcription factor AP2-beta (TFAP2B; MIM 601601), and transcription factor AP2-gamma (TFAP2C; MIM 601602) via its BTB domain and acts as a strong repressor of *TFAP2A* transcriptional activity [12,19]. Therefore, KCTD1 functions as a potent transcriptional repressor of TFAP2A and a negative regulator of WNT/β-catenin signaling, which is important for embryonic development and homeostatic self-renewal in tissue regeneration and repair [8,21]. Of note, KCTD1 has also been reported to function as a suppressor of the hedgehog pathway [22].

The BTB domain of KCTD1 is located at the N-terminus of the protein. It generally forms a close pentameric structure when it binds to its functional partner [23]. The BTB domain has been highly conserved throughout evolution from *Drosophila* to mammals. Interestingly, the 257 amino acid mouse Kctd1 and human KCTD1 are identical, while only a single amino acid difference is found in rat [23]. The frequent association of KCTD1 with putative transcriptional regulators containing zinc finger motifs suggests that it plays an important role in transcriptional regulation [12,19,21,23,24].

Here, we report a novel variant and a rare variant of *KCTD1* in nine patients from two unrelated families with isolated dental anomalies including taurodontism, unseparated roots, long roots, tooth agenesis, a supernumerary tooth, torus palatinus, and torus mandibularis. Maldevelopment of roots appeared to be a common finding. Our study sheds light on a novel understanding of the influence of *KCTD1* variants on odontogenesis.

## 2. Results

### 2.1. Clinical Description of Patients

Nine patients from two unrelated families were found to have taurodontism, unseparated roots, generalized thin and tapered roots, long roots, shortened roots, tooth agenesis, microdontia, a supernumerary tooth, torus palatinus, and torus mandibularis (Table 1, Figure 1, Figure 2 and Figure 3).

### 2.2. Whole Exome Sequencing and Bioinformatic Analysis

Whole exome sequencing was performed on all 362 affected individuals. We initially focused on unrelated individuals that carried rare variants in the *KCTD1* gene. This assessment identified two potentially pathogenic variants in *KCTD1:* one rare missense variant (c.722G>A; p.Arg241Gln) and one novel missense variant (c.727C>T; p.Pro243Ser). Both individuals presented with similar isolated dental anomalies with or without torus palatinus and mandibularis. Subsequent Sanger sequencing of amplicons showed that all affected family members from both families (nine patients total in the two families) carried the respective heterozygous *KCTD1* variant, while all the unaffected members of the two families did not, strongly supporting a genotype–phenotype correlation (Figure 1 and Figure 4). None of the nine patients had any clinical findings consistent with Scalp-Ear-Nipple syndrome (Table 1, Figure 1, Figure 2 and Figure 3).

In order to rule out other known genetic causes of dental anomalies, we screened the whole exome sequencing results of all nine patients for other rare variants with allele frequencies < 0.0003, focusing on genes including *WNT10A, WNT10B, LRP5, LRP6, PAX9, AXIN2, MSX1, WLS, BMP4, KDF1, ATF1, DUSP10, EDA, EDAR, EDARADD, GREM2, TFAP2B, TSPEAR, PITX2, EVC, EVC2, COL1A2, ANTXR1, FGF10, KREMEN1, CASC8,* and *SMOC2* [25,26,27,28]. This assessment identified only a single rare variant in *LRP5* (chr11:g.68201132G>A; NM_001291902.2: c.2083G>A; NP_001278831.1: p.Gly695Arg; rs370744430) in patient 7 who had unseparated roots of teeth 17 and 27, agenesis of teeth 25, 36, 38, and 47, microdontia of teeth 12 and 22, and a torus palatinus. The allele frequency of this variant is 0.00000716 (gnomAD database, v2.1). In a multiple sequence alignment of LRP5 proteins from 494 species of vertebrates, the amino acid residue Gly695 is found in 117 species of vertebrates, suggesting it is not well conserved. The *LRP5* variant c.2083G>A; p.Gly695Arg is predicted to be a polymorphism (prob = 0.860781214034341) and benign (0.003) by MutationTaster and PolyPhen-2, respectively. The Combined Annotation Dependent Depletion (CADD) and Deleterious Annotation of genetic variants using Neural Networks (DANN) scores for this variant are 21.8 and 0.9059, respectively. Since genetic variants in *LRP5* have been reported to be associated with tooth agenesis, taurodontism, and oral exostoses, it is possible that this *LRP5* variant contributes to the dental phenotypes and torus palatinus in patient 7. However, the variant was not found in other affected family members, ruling it out as the primary causal variant in this family.

### 2.3. p.Arg241Gln Variant

A rare variant, c.722G>A; p.Arg241Gln, was identified in patients 1, 2, 3, 4, 5, and 6 of family 1, who had tooth agenesis, taurodontism, unseparated roots of permanent molars, long roots, root dilaceration, torus palatinus, and torus mandibularis (Figure 4a). According to gnomAD, this variant is rare, with an allele frequency of 0.00001193. This variant has not been reported in the South Asian and East Asian populations. According to the T-REx database, this allele was found in 1 out of 2184 alleles (allele frequency = T:0.00045788 or 0.04%) in Thai subjects. A multiple sequence alignment showed that the amino acid residue Arg241 is conserved across 438 out of 476 species of vertebrates (Figure 5). This variant is predicted to be disease causing by MutationTaster (prob = 0.999946005902276) (https://www.mutationtaster.org; accessed on 24 September 2023), probably damaging by PolyPhen-2 (0.997) (http://genetics.bwh.harvard.edu/pph2/; accessed on 24 September 2023), and damaging by SIFT (0.01) (https://bio.tools/sift; accessed on 24 September 2023). The DANN score of this variant (https://cbcl.ics.uci.edu/public_data/DANN/; accessed on 24 September 2023) is 0.9988, which suggests the association between the variant and disease risk (https://varsome.com/about/resources/germline-implementation/ accessed on 24 September 2023). The CADD of this variant (https://cadd.gs.washington.edu; accessed on 24 September 2023) is 23.5, suggesting that this variant is predicted to be among the 1.0% most deleterious possible substitutions in the human genome (https://genome.ucsc.edu/ accessed on 24 September 2023).

### 2.4. p.Pro243Ser Variant

The heterozygous missense variant c.727C>T; p.Pro243Ser was identified in patients 7, 8 and 9, of family 2 who had tooth agenesis, taurodontism, a supernumerary tooth, unseparated roots of permanent molars, and a torus palatinus (Figure 4b). This variant is not reported in gnomAD (https://gnomad.broadinstitute.org; accessed on 24 September 2023), LOVD (https://www.lovd.nl; accessed on 24 September 2023), HGMD (https://www.hgmd.cf.ac.uk/ac/index.php; accessed on 24 September 2023), or the Thai reference exome variant database (T-REx) [29]; therefore, it is considered novel. The amino acid residue Pro 243 is highly conserved, being present in 289 out of 476 species of vertebrates, a selection of which are shown in Figure 5. This variant is predicted to be a polymorphism by MutationTaster (0.931756239937238) (https://www.mutationtaster.org; accessed on 24 September 2023), benign by PolyPhen-2 (0.007) (http://genetics.bwh.harvard.edu/pph2/; accessed on 24 September 2023), and tolerated by Sorting Intolerant From Tolerant (0.1) (SIFT; https://bio.tools/sift; accessed on 24 September 2023). The CADD (https://cadd.gs.washington.edu; accessed on 24 September 2023) and DANN (https://cbcl.ics.uci.edu/public_data/DANN/; accessed on 24 September 2023) scores for this variant are 16.16 and 0.9425, respectively, suggesting it may be benign (https://varsome.com/about/resources/germline-implementation/ accessed on 24 September 2023). Neither variant has any predicted cryptic impact on *KCTD1* mRNA splicing, as determined using SpliceAI and Pangolin (https://spliceailookup.broadinstitute.org/ accessed on 24 September 2023).

### 2.5. Functional Studies

KCTD1 functions as an inhibitor of the WNT/β-catenin signaling pathway, and cytoplasmic β-catenin levels reflect the WNT signaling activity [30]. Immunoblotting was conducted to investigate the impact of p.Arg241Gln and p.Pro243Ser variants on the expression of KCTD1 and β-catenin. The results demonstrated that both p.Arg241Gln and p.Pro243Ser proteins were expressed in the cells, with band intensities comparable to the wild-type protein. Interestingly, p.Arg241Gln led to a significant reduction in β-catenin levels compared to the WT. This indicates that the alteration in the p.Arg241Gln variant decreased WNT signaling activity. In contrast, the p.Pro243Ser variant produced a higher level of β-catenin, but it was not significantly different from that of wild type (Figure 6a,b; Appendix A; Appendix A).

### 2.6. Kctd1 Expression in Mouse Tooth Development

Expression of Kctd1 in the developing teeth of mice was assessed by immunofluorescence. Expression was not evident in mouse tooth germs at embryonic day E12.5 (Figure 7a) but was detectable in tooth epithelium at E13.5 and E14.5 (Figure 7b,c). Kctd1 is not expressed in dental epithelium at the bud stage, while it is expressed at the cap stage. Kctd1 expression was also detected in ameloblasts and Hertwig epithelial root sheath at postnatal day (P) 10 (Figure 7d,e). At all stages, Kctd1 expression was detected in the oral epithelium and then increasingly in mesenchymal cells.

### 2.7. The Structure of KCTD1

The positions of the mutated residues were inspected in the human KCTD1 structure (Protein Databank, PDB, https://www.rcsb.org/structure/6s4l, accession 6S4L; accessed on 24 September 2023). The previously reported KCTD1 variants are found exclusively in the N-terminal disordered region and BTB domain (Figure 8a). The variants reported in this study p.Arg241Gln and p.Pro243Ser are located adjacent to the C-terminal H1 tetramerization domain and are neighboring other highly conserved residues in the turn between the C-terminal domain and the extended tail (Figure 8b,c). Both substitutions are likely to increase the flexibility of this region, particularly conversion from Pro, which has limited backbone flexibility, to Ser at 243, so they may affect the angle at which the C-terminal tail is extended. The proximity of the two variants and the high degree of conservation of these residues suggests that they could have similar impacts on protein function, thus explaining the similar dental anomalies.

## 3. Discussion

### 3.1. KCTD1 Variants Are Associated with Isolated Dental Anomalies

Whole exome sequencing of 362 patients with dental anomalies identified a rare (p.Arg241Gln) variant and a novel variant (p.Pro243Ser) of *KCTD1,* both of which completely segregated with the dental presentations in the respective families (a total of nine individuals) supporting a causal relationship. The isolated dental anomalies observed in our patients consist of taurodontism, unseparated roots, long roots, tooth agenesis, a supernumerary tooth, torus palatinus, and torus mandibularis. Both *KCTD1* variants found in our patients, p.Arg241Gln and p.Pro243Ser, reside in close proximity to each other in the C-terminal domain of the protein and both impact highly conserved residues. This is in contrast with all previously reported variants in *KCTD1* associated with Scalp-Ear-Nipple syndrome, which reside in the N-terminal disordered region or in the BTB domain, suggesting a possible genotype–phenotype correlation. Until now, no pathological variants have been reported in the C-terminal end of KCTD1 protein [10,11,12,13,14]. More patients with isolated dental anomalies and KCTD1 variants in the C-terminal domain would confirm a causal relationship.

### 3.2. Possible Impacts of KCTD1 Variants on Their Functions

The Arg241Gln variant is predicted to be deleterious or damaging by multiple algorithms. In addition, this is supported by our functional study, which demonstrated that the alteration in Arg241Gln protein had a significant effect on β-catenin levels. Based on the protein structure, Arg241 is located at the hinge point between a highly compact region and the extended unstructured tail. The change to glutamine at this position will decrease the overall positive charge and the size of the side chain, which may increase flexibility of the C-terminal end. The change in the charge distribution could potentially affect the interaction with interacting proteins, including those involved in posttranslational modification of the C-terminal tail. The deleterious effect of this mutant protein model is in support of its functional study at the cellular level.

Although the p.Pro243Ser variant is computationally predicted to be benign and its functional study did not show that it significantly affects β-catenin levels, its novelty, high degree of conservation, segregation with the phenotypes, and proximity to Arg241 still warrant its consideration as potentially pathogenic. The p.Pro243Ser change may similarly affect the angle at which the C-terminal tail extends since proline in this position would potentially restrict the main chain torsion angles due to the connection of its side chain to its amino nitrogen. Such changes in the C-terminal tail angle could impact the interaction of the tail with other proteins. In this regard, the KCTD1 protein has been identified as a substrate for SUMOylation, an important post-translational modification that regulates protein function in eukaryotes [31,32]. A SUMOplot (https://www.abcepta.com/sumoplot; accessed on 24 September 2023) analysis identified a predicted consensus sequence for SUMOylation, ψKXE (with SUMOylation at K252), in the C-terminal tail of KCTD1. Notably, SUMOlyation plays important roles in tooth development [33]. Therefore, it is possible that the variants in our patients interfere with the SUMOlyation of KCTD1, although follow up functional studies would be required to test this. It should be noted that the p.Pro243Ser variant also introduces a new potential phosphorylation or O-GlcNAc site, raising an alternate possible mechanism to disrupt partner protein interactions and normal protein function. Therefore, it is possible that the altered KCTD1 protein may have an effect on tooth development that is not related to WNT signaling. It is nevertheless important to note that dental abnormalities are not life threatening and do not result in severe disability. Most patients with dental anomalies, especially those with root maldevelopments, do not even know they have them. It is therefore our considered opinion that mutation algorithms may not accurately predict the effects of all genetic variants underlying relatively benign dental variations.

### 3.3. KCTD1 Variants, Tooth Agenesis, Microdontia, and Supernumerary Tooth Formation

The role of KCTD1 is to repress canonical WNT/β-catenin signaling by enhancing β-catenin degradation [20]. Disruption of KCTD1 function as a result of genetic variants might result in upregulation of WNT signaling in the dental epithelium, with secondary impact on the neural crest-derived mesenchyme and the genes specifying incisor and molar fate, which could explain agenesis and formation of a supernumerary maxillary tooth in the same individual [34,35,36]. In line with a causal role for *KCTD1* variants in various dental anomalies, our immunofluorescence investigation showed that detectable Kctd1 expression begins at the cap stage.

*Tfap2a* and *Tfap2b* have been demonstrated to have important roles in tooth patterning [37]. As previously mentioned, KCTD1 has been reported to interact with TFAP2A, TFAP2B, and TFAP2C [12]. TFAP2A inhibits the WNT/β-catenin signaling by associating with APC protein. Subsequently, the TFAP2A/APC/β-catenin complex induces the nuclear β-catenin into an inactive state and interrupts its binding to T-cell factor/lymphoid enhancer factor (TCF/LEF) transcription factors. Thus, genetic variants in *KCTD1* might disrupt WNT/β-catenin signaling via its interaction with TFAP2A and TFAP2B and subsequent tooth development. This is supported by the findings of isolated dental anomalies, including tooth agenesis, microdontia, supernumerary teeth, and root maldevelopments, in patients with *TFAP2B* variants [38].

SHH also modulates the growth and shape of the tooth, and aberrant Shh signaling during early tooth development in mice results in tooth malformations [39,40]. KCTD1 has also been reported to function as a suppressor of the hedgehog pathway [22]. So, it remains possible that the tooth agenesis, microdontia, supernumerary tooth, torus palatinus, and torus mandibularis found in our patients’ results, at least in part, are from the impact of the KCTD1 variants on SHH signaling.

### 3.4. KCTD1 Variants and Root Maldevelopment

Root maldevelopments including taurodontism, unseparated roots of molars, shortened roots, and generalized long roots were found in all nine patients with *KCTD1* variants. This implies an important role of *KCTD1* in development of roots. Taurodontism is caused by decreased proliferation of Hertwig epithelial root sheath cells at the initial stage of root development and excessive proliferation of adjacent dental mesenchyme cells in dental papilla during root development [41]. The fundamental defects of the root maldevelopments appear to relate to disruptive root lengthening and formation of root furcations [41]. The elongation and invagination of Hertwig epithelial root sheath cells into the underlying mesenchyme, which determines the length and number of roots, is also strongly dependent on WNT/β-catenin signaling [41,42,43,44,45]. In support of this, taurodontism or unseparated roots of molars is seen in *Wnt10a* knockout mice [46,47] and in most patients with genetic variants in WNT/β-catenin signaling such as *WNT10A*, *WNT10B*, *LRP4*, *LRP5*, *LRP6*, *DKK1*, *WLS*, and *DLX3* [48,49,50].

SHH and BMP signaling also play crucial roles in root development [42,51], with genetic variants in *BMP4* having been implicated in tooth agenesis, root maldevelopment, and oral exostoses, similar to the features found in our patients with *KCTD1* variants [26]. Therefore, in addition to aberrant WNT and SHH signaling, the root maldevelopment in our patients with *KCTD1* variants might have been the effects of disruptive BMP signaling as well.

Taken together, it is hypothesized that root maldevelopments in patients 1–9 were the consequences of *KCTD1* variants that affected *KCTD1* protein function and subsequently led to abnormal root development. Our finding of immunofluorescence study of Kctd1 expression in the Hertwig epithelial root sheath at postnatal days supports the hypothesis.

### 3.5. KCTD1 Variants, Torus Palatinus, and Torus Mandibularis

The finding of torus palatinus and torus mandibularis in patients 2 and 7 can also be explained by aberrant WNT/β-catenin and BMP signaling, as pathogenic variants in *LRP5*, *BMP4*, *LRP6*, *LRP4*, *DKK1*, and *WLS* are implicated in the formation of these tori [26,27,28,52]. Patient 7 also carried an *LRP5* variant, which might contribute to the formation of the torus palatinus, in conjunction with the *KCTD1* variant. Dysregulation of BMP signaling similarly can lead to various bone disorders, including abnormal bone mass [53], and might contribute to the formation of the tori in our patients with *KCTD1* variants.

## 4. Materials and Methods

### 4.1. Ethic Statement

This study was conducted with informed consent from all participants or their parents and was approved by the Human Experimentation Committee of the Faculty of Dentistry, Chiang Mai University (certificate of approval number 71/2020) and is in accordance with the ethical standards of the Declaration of Helsinki.

### 4.2. Patient Recruitment and Clinical Investigations

Inclusion criteria for patient recruitment were patients with isolated dental anomalies, including tooth agenesis, microdontia, macrodontia, supernumerary teeth, odontomas, talon cusps, enamel hypoplasia, tooth fusion, dens evaginatus, failure of tooth eruption, and root anomalies. Oral and radiographic examinations were performed on the cohort of 362 patients who came to The Pediatric Dental Clinic, Faculty of Dentistry, Chiang Mai University (Figure 9). The participants included 196 (54.14%) females and 166 (45.86%) males. The genetic studies of this cohort have been published [6,26,50].

### 4.3. Whole Exome Sequencing

Depending on the availability, either saliva or blood from the patients was collected as a source of genomic DNA to be sequenced. The saliva collection procedure follows the protocols outlined in the Oragene DNA OG-575 kit (DNA Genotek Inc., Ottawa, ON, Canada), and genomic DNA was extracted from the saliva using an ethanol precipitation protocol and the prepIT L2P reagent. The blood collection procedure utilizes 4 mL of blood in BD Vacutainer® EDTA tubes (10.0 mL K2E, containing EDTA 18.0 mg) (BD-Plymouth, PL6 7BP, UK), and genomic DNA was extracted from the blood using protocol 131 of the QuickGene DNA whole blood kit (Kurabo Industries Limited, Osaka, Japan).

Whole exome sequencing (WES) was performed on the genomic DNA of all patients. The SureSelect exome capture library (Agilent Technologies, Santa Clara, CA, USA) was used to target all coding exons and UTRs of the human genome. The output sequencing reads in the FASTQ format were aligned using BWA-MEM to the reference sequence, hg19/GRCh37. The duplication reads were marked with GATK-MarkDuplicate utility. Following the GATK best practices, base quality score recalibration was used prior to HaplotypeCaller to produce each patient’s list of variants (VCF file format). GATK GenotypeGVCF was used to call genotype from all individuals. The resulting variants including SNVs and INDELs were combined with those variants from the in-house cohort (799 samples), producing a combined VCF of 1161 samples. We filtered these variants again using GATK-VQSR and fed the passed variants to Variant Effect Predictor (VEP) build 110 with dbNSFP 4.4a to annotate them with the predicted variant functional effects. At this stage, there were over 32 million variants that were subject to the VEP prediction. To further narrow down the huge search space, we exercised the predisposing gene list hypothesis and adopted hard-filtering criteria such as considering only those with moderate to high impact on the predicted functions, which were really rare or novel (MAF < 0.0009999 in gnomADe, gnomADg, 1000 Genomes databases and MAF < 0.01 from the Thai specific variant database (T-REx)). The hard-filtering criteria reduced the number of search space down to approximately 4000 variants (Figure 10). We found the candidate variants in the *KCTD1* gene. Sanger sequencing was performed to confirm the variants. The sequence primers used were as follows: Exon5, forward: 5′-TTGCTGTCCCAACTGCACATA-3′; reverse: 5′-ACATGGGTGCTGGATGAGATG-3′.

### 4.4. Assessment of Protein Sequence and Structure

The human sequence was aligned with 476 vertebrate KCTD1 protein sequences with Clustal Omega, (https://www.ebi.ac.uk/Tools/msa/clustalo/; accessed on 24 September 2023) to generate a multiple sequence alignment and evaluate the conservation of the mutated amino acids. The context of the mutations in the KCTD1 protein structure was analyzed based on the crystal structure of the full-length pentamer structure of KCTD1 (PDB entry 6S4L—doi: 10.2210/pdb6S4L/pdb). The protein models carrying mutations were made by simply changing the amino acids and selecting the most favorable rotamers, and wild-type and mutant structures were visualized in PyMol (version 2.5.0, Schrödinger LLC, New York, NY, USA).

### 4.5. Functional Studies

Recombinant Plasmids

Recombinant plasmids were purchased from GeneScript (Piscataway, NJ, USA). The 3FLAG-tagged pCMV expression vector contained wild-type KCTD1 (CCDS11888.1) or KCTD1 harboring either the Arg241Gln or Pro243Ser variant. The nucleotide sequences of all constructed plasmids were verified after the construction.

Cell Transfection

Human embryonic kidney 293 (HEK293, CRL-1573) were seeded in 24-well plates and cultured in Dulbecco’s modified eagle medium (DMEM, Gibco BRL, Carlsbad, CA, USA) with supplement of 10% fetal bovine serum (HyClone, Logan, UT, USA), 1% L-glutamine, 100 U/mL penicillin, and 100 µg/mL streptomycin (Gibco BRL) in a 5% CO_2_ humidified atmosphere at 37 °C. After 24 h of incubation, cells were transfected by using Lipofectamine 3000 (Invitrogen, Carlsbad, CA, USA) according to the manufacturer’s instruction.

SDS-PAGE and Immunoblotting

The whole cell lysate of transfected HEK293 was subjected to sodium dodecyl-sulfate polyacrylamide gel electrophoresis (SDS-PAGE) and subsequent immunoblot analysis. Blots were probed with primary antibodies (mouse anti-FLAG (Merck, St. Louis, MO, USA, Cat. No. F1804-200UG), mouse anti-β-tubulin (BioLegend, Inc., San Diego, CA, USA, Cat. No. 605102), and rabbit anti-β-catenin (Sigma-Aldrich, St. Louis, MO, USA, Cat. No. C2206) and secondary antibodies (goat anti-mouse IgG and goat anti-rabbit IgG HRP conjugated IgG (R&D Systems, Inc., Minneapolis, MN, USA, Cat. No. HAF007 and HAF008)). The bands were visualized using the superSignal^TM^ West Femto maximum sensitivity substrate, and images were captured using the Amersham Imager 600 (GE Healthcare, Chicago, IL, USA). Immunoblotting was performed in duplicates and carried out for two independent experiments. The intensity of the bands was determined using ImageJ 1.54 software.

Statistical Analysis

Data are presented as means ± standard deviations (SD). GraphPad Prism 9.4.1 was used for statistical analysis of band intensities (n = 4) using the Mann–Whitney U test to compare differences between two independent groups. A *p*-value ≤ 0.05 was considered to be statistically significant.

### 4.6. Immunohistochemical Study

CD-1 strain mice were used in this study. Embryo and newborn mouse heads were fixed in 4% buffered paraformaldehyde, wax embedded, and serially sectioned at 7 µm. Sections were incubated with the antibodies against *Kctd1* (Invitrogen; PA5-24877). The tyramide signal amplification system was used (Parkin Elmer Life Science, Waltham, MA, USA) to detect the *Kctd1* antibody.

## 5. Limitations of the Study

Cone beam computed tomography would have been preferable to further characterize the root maldevelopment seen in all patients with the *KCTD1* variant, but this technology was not clinically available at the times of assessment. Functional studies of p.Pro243Ser did not show significant effects on β-catenin levels. However, our SUMOplot analysis identified a predicted consensus sequence for SUMOylation, ψKXE (with SUMOylation at K252), in the C-terminal tail of KCTD1. Therefore, functional studies to test the effect of the p.Pro243Ser variant on SUMOylation may have supported the effects of this variant on tooth development. Lastly, functional studies of the effects of both *KCTD1* variants on SHH signaling could be performed in the future to directly test the effects of these variants on this pathway that also plays a critical role in tooth development.

## 6. Conclusions

In summary, we present evidence to support the contribution of *KCTD1* variants to the presentation of isolated dental anomalies and oral exostoses in our two families because (1) both variants are rare (p.Arg241Gln) or novel (p.Pro243Ser); (2) the variants fully segregated with the phenotypes in each family; (3) previously reported patients with *KCTD1* mutation-associated Scalp-Ear-Nipple syndrome were reported to have dental anomalies, indicating the link between *KCTD1* and tooth development; (4) the location of our identified variants is in the C-terminal end of KCTD1, compared to those of patients with Scalp-Ear-Nipple syndrome, who had variants located in the N-terminal BTB domain; and (5) no other rare variants in the known dental anomalies-associated genes were identified to cosegregate with dental anomalies in the families. This study demonstrates for the first time that genetic variants in *KCTD1* are associated with isolated dental anomalies. This is supported by the functional studies, which showed that the p.Arg241Gln KCTD1 variant affected β-catenin levels, an effector of Wnt/β-catenin signaling. Both variants (p.Arg241Gln and p.Pro243Ser) are located in the C-terminal tail of the protein, suggesting a possible genotype–phenotype correlation. Further studies addressing the impact of the variants on protein–protein interactions, protein stability, and SUMOylation, and subsequent aberrant WNT-SHH-BMP signaling are needed.

## Figures and Tables

**Figure 1 ijms-25-05179-f001:**
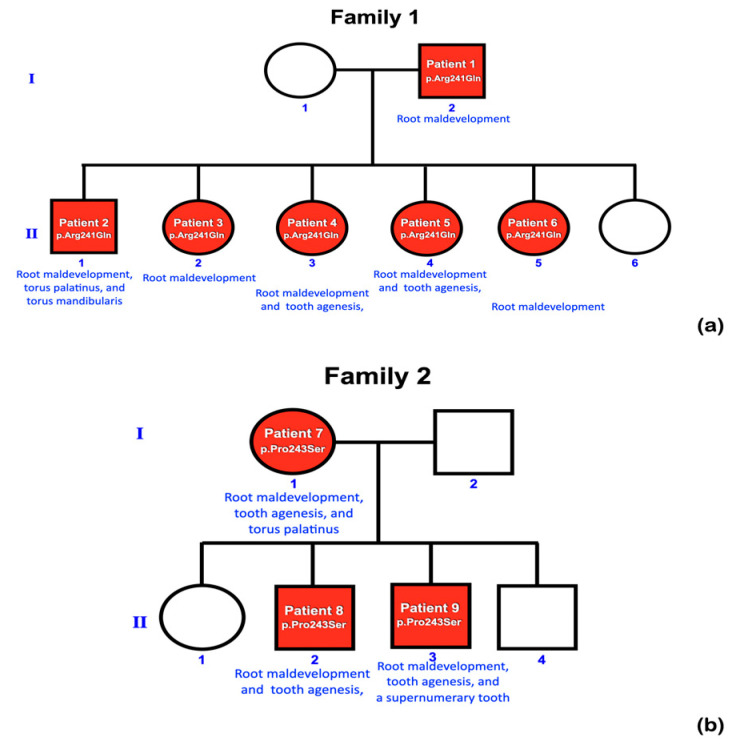
Pedigrees of (**a**) family 1 and (**b**) family 2. Root maldevelopment is a consistent finding in patients with *KCTD1* variants.

**Figure 2 ijms-25-05179-f002:**
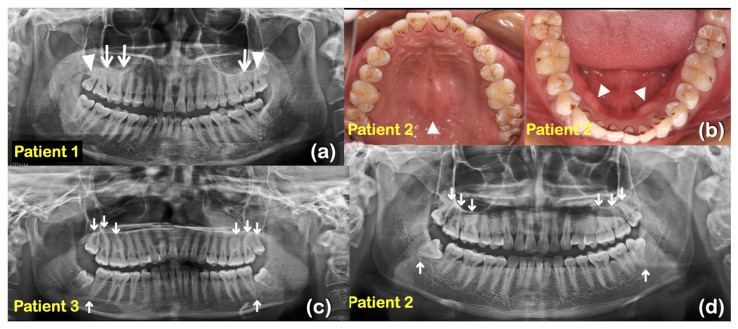
Family 1 (**a**) Patient 1 at age 48 years. Panoramic radiograph showing taurodontism of teeth 16, 17, and 27 (arrows). Teeth 18 and 28 have unseparated roots (arrowheads). Generalized long roots. (**b**) Patient 2 at age 20 years. Maxillary teeth and torus palatinus (arrowhead). Mandibular teeth and torus mandibularis (arrowheads). (**c**) Patient 3 at age 17 years. Panoramic radiograph showing taurodontism of teeth 16, 17, 18, 26, 27, 28, 37, and 47 (arrows). (**d**) Patient 2 at age 20 years. Panoramic radiograph showing taurodontism of teeth 16, 17, 18, 26, 27, 28, 38, and 48 (arrows).

**Figure 3 ijms-25-05179-f003:**
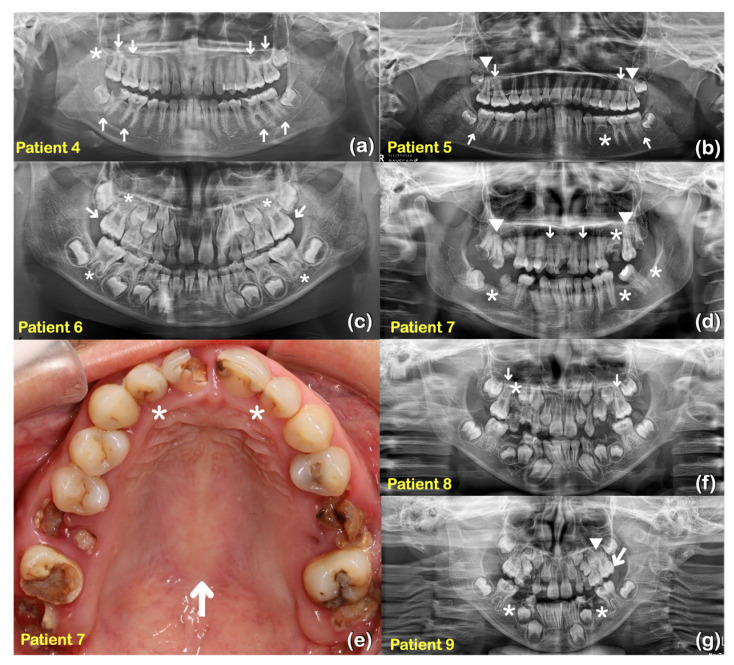
Family 1 (**a**) Patient 4 at age 14 years. Panoramic radiograph showing taurodontism of teeth 16, 17, 26, 27, 36, 37, 46, and 47 (arrows). Generalized long roots. Agenesis of tooth 18 (asterisk). (**b**) Patient 5 at age 12 years. Panoramic radiograph showing taurodontism of teeth 16, 26, 37, and 47 (arrows). Unseparated roots of teeth 17 and 27 (arrowheads). Generalized thin and tapered roots. Agenesis of tooth 35 (asterisk). (**c**) Patient 6 at age 7 years. Panoramic radiograph showing mixed dentition and taurodontism of teeth 16 and 26 (arrows). Shortened roots of teeth 16, 26, 36, and 46 (asterisks). Family 2 (**d**) Patient 7 at age 28 years. Panoramic radiograph showing unseparated roots of teeth 17 and 27 (arrowheads). Agenesis of teeth 25, 36, 38, and 47 (asterisks). Microdontia of teeth 12 and 22 (arrows). (**e**) Patient 7. Microdontia of teeth 12 and 22 (asterisks). Torus palatinus (arrow). (**f**) Patient 8 at age 10 years. Panoramic radiograph showing mixed dentition. Taurodontism of teeth 16 and 26 (arrows). Agenesis of tooth 15 (asterisk). (**g**) Patient 9 at age 7 years. Panoramic radiograph showing mixed dentition, taurodontism of tooth 26 (arrow). Agenesis of teeth 35 and 45 (asterisks). A supernumerary maxillary tooth (arrowhead).

**Figure 4 ijms-25-05179-f004:**
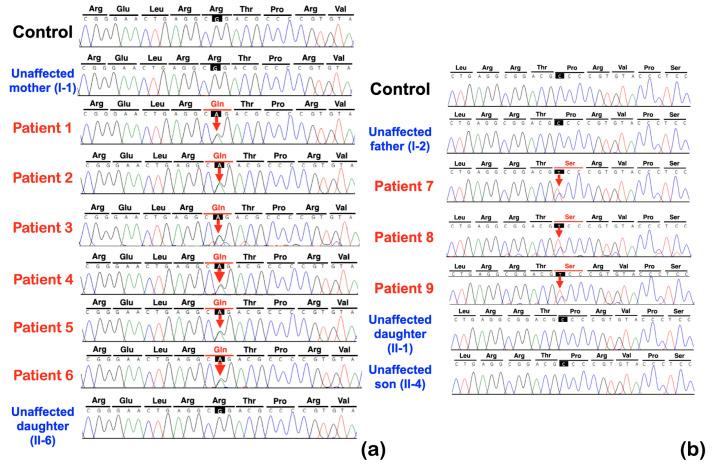
Sequence chromatograms of *KCTD1* variants. Family 1. (**a**) The heterozygous missense variant c.722G>A (p.Arg241Gln) is identified in patients 1–6 but not in the control and unaffected mother (I-1) and unaffected daughter (II-6). Family 2. (**b**) The heterozygous missense variant c.727C>T (p.Pro243Ser) is identified in patients 7–9 but not in the control, unaffected father (I-2), unaffected daughter (II-1), and unaffected son (II-4).

**Figure 5 ijms-25-05179-f005:**
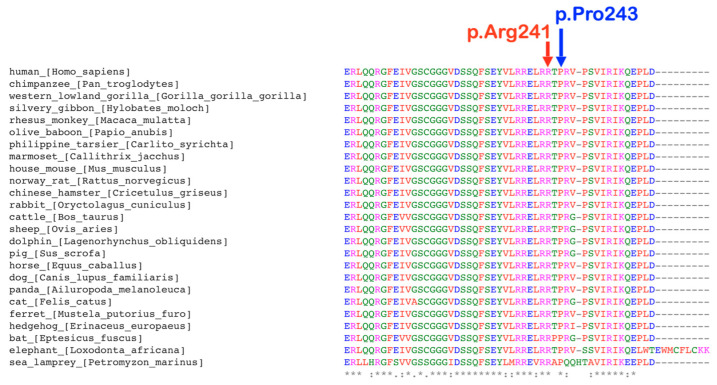
Conservation of amino acids. The amino acids Pro243 and Arg 241 are highly conserved.

**Figure 6 ijms-25-05179-f006:**
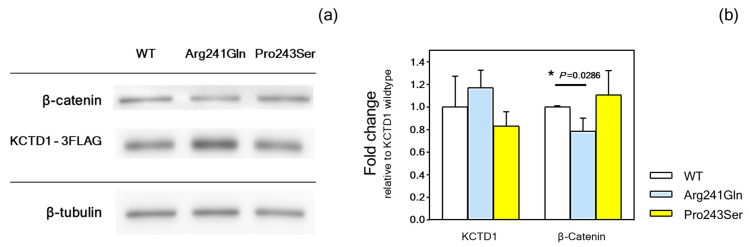
Immunoblotting of KCTD1. (**a**) Image of typical immunoblot bands of wild-type KCTD- WT, p.Arg241Gln, and p.Pro243Ser variants, along with β-catenin and β-tubulin. (**b**) The fold change in the intensity of the bands relative to the wild-type KCTD1 was compared and shown in the bar chart (means and standard deviations of four experiments). Statistical significance is indicated by *.

**Figure 7 ijms-25-05179-f007:**
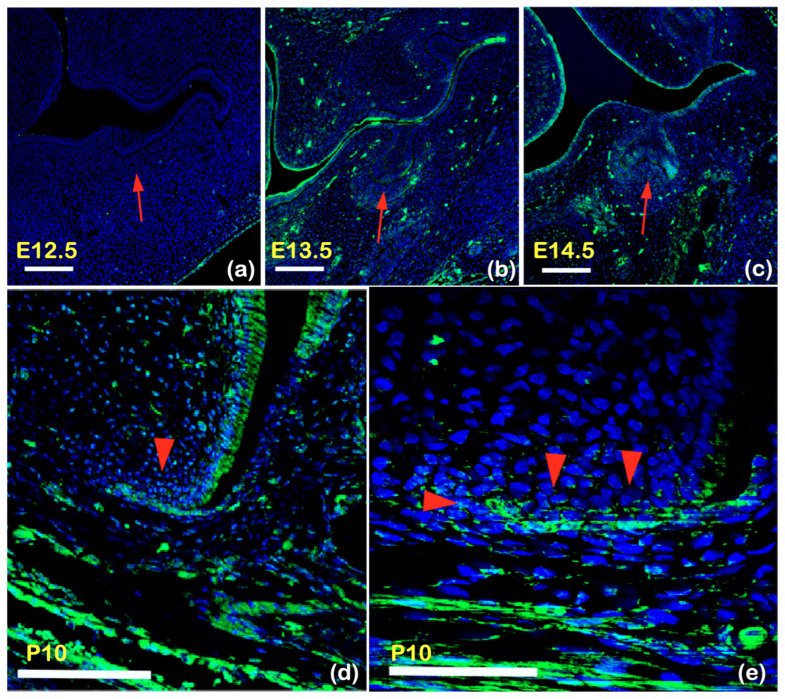
Frontal section showing immunohistochemistry of Kctd1 in wild-type mouse at E12.5 (**a**), E13.5 (**b**), E14.5 (**c**), and P10 (**d**). (**a**) Immunohistochemical study of Kctd1 expression of frontal sections of wild-type mouse embryo during early mouse tooth development. No expression of Kctd1 is observed in early bud stage of tooth germ at embryonic day (E) 12.5 (E12.5) (arrow). (**b**,**c**) Kctd1 is expressed in dental epithelium (arrows) at (**b**) late bud stage (E13.5) and (**c**) cap stage (E14.5). (**d**,**e**) Kctd1 expression is found in ameloblasts and HERS at postnatal day (P) 10 (P10) (arrowheads), suggesting its important role in root development. (**e**) High magnification arrowhead indicated in (**d**). Scale bars; 100 μm (**a**–**d**) and 50 μm (**e**). Arrows and arrowhead indicate tooth germ and Hertwig epithelial root sheath, respectively. Kctd1 is not expressed in dental epithelium at the bud stage, while it is expressed at the cap stage. Kctd1 is highly expressed in Hertwig epithelial root sheath.

**Figure 8 ijms-25-05179-f008:**
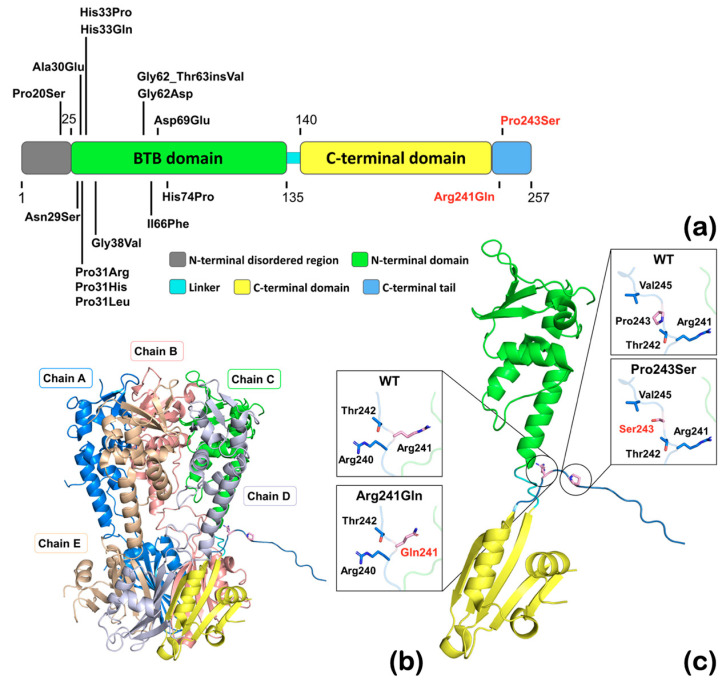
The structure of KCTD1 protein showing the positions of the Arg241Gln, Pro243Ser, and previously reported mutations. (**a**) A schematic map of the primary structure of human KCTD1 based on the structure of KCTD1 showing the mutation locations. Structural domains are coded by color as indicated. The starting and ending positions of each structural domain are marked. The residue numbering is based on accession number NCBI: NP_001129677.1. Note that the previously reported KCTD1 variants (in black) are found exclusively in the N-terminal disordered region and BTB domain. The KCTD1 variants reported in this study (in red) are found in the C-terminal tail. (**b**) The full-length structure of KCTD1 (PDB entry 6S4L—doi: 10.2210/pdb6S4L/pdb) is a pentamer and covers, with the exception of 25 N-terminal and 4 C-terminal amino acid residues, the entire protein. (**c**) The monomer of KCTD1 (PDB entry 6S4L—doi: 10.2210/pdb6S4L/pdb) with the expanded view of mutated and neighboring amino acid side chains shown in stick. Both mutations change amino acids at the turn from the C-terminal domain to the extended flexible tail and change from larger to smaller, more flexible amino acid sidechains, thereby likely changing the position of the extended C-terminus.

**Figure 9 ijms-25-05179-f009:**
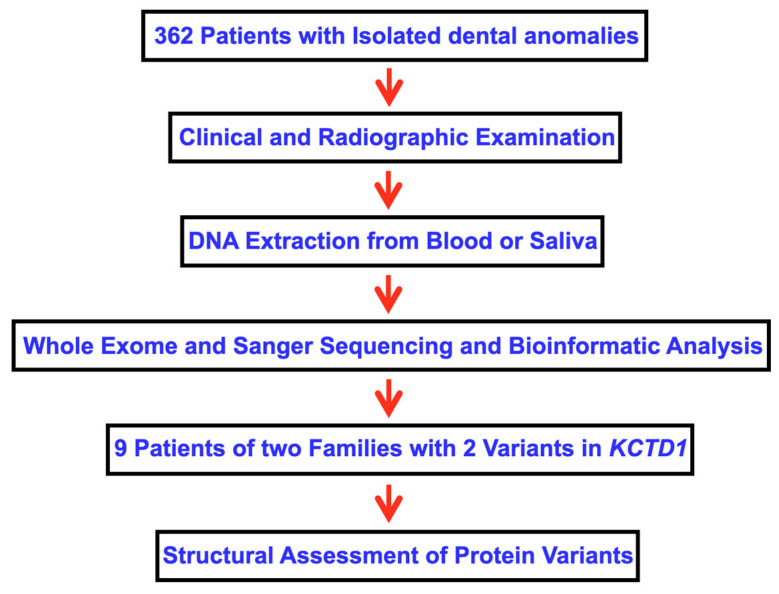
Flowchart describing the methodology of this study.

**Figure 10 ijms-25-05179-f010:**
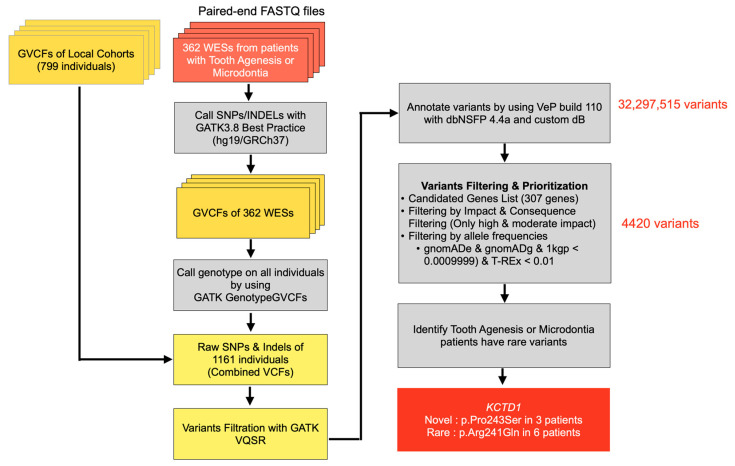
Flowchart showing steps in bioinformatic analysis.

**Table 1 ijms-25-05179-t001:** Patients with *KCTD1* variants and their dental phenotypes.

Patients Genders	Families	Phenotypes	*KCTD1* Variants
1 (I-2; Male)	1	Taurodontism of teeth 16, 17, and 27; 18, 28 unseparated root; generalized long roots	c.722G>A p.Arg241Gln chr18-24035759-C-T rs776466895 MAF = 0.00001193 DANN score = 0.9988 CADD score = 23.5
2 (II-1; Male)		Taurodontism of teeth 16, 17, 18, 26, 27, 28, 38, and 48; torus palatinus and torus mandibularis	
3 (II-2; Female)		Taurodontism of teeth 16, 17, 18, 26, 27, 28, 37, and 47	
4 (II-3; Female)		Taurodontism of teeth 16, 17, 26, 27, 36, 37, 46, and 47; generalized long roots; agenesis of tooth 18	
5 (II-4; Female)		Taurodontism of teeth 16, 26, 37, and 47; unseparated roots of teeth 17 and 27; generalized thin and tapered roots; agenesis of tooth 35	
6 (II-5; Female)		Taurodontism of teeth 16 and 26; shortened roots of teeth 16, 26, 36, and 46	
7 (I-1; Female)	2	Unseparated roots of teeth 17, 27; agenesis of teeth 25, 36, 38, and 47; microdontia of teeth 12, 22; torus palatinus	c.727C>T; p.Pro243Ser chr18-24035754-G-A NOVEL DANN score = 0.9425 CADD score = 16.16
8 (II-2; Male)		Taurodontism of teeth 16, and 26; agenesis of tooth 15	
9 (II-3; Male)		Taurodontism of tooth 26; agenesis of teeth 35 and 45; a supernumerary tooth	

## Data Availability

Data are contained within the article and Appendix A.

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
