# Peer review of "Genetic Variants in KCTD1 Are Associated with Isolated Dental Anomalies"

_ijms, 2024, doi:10.3390/ijms25105179_

Round 1

Reviewer 1 Report

Comments and Suggestions for Authors

The authors made quite an effort to conduct genetical study of finding association between genetic variants in KCTD1 and isolated dental anomalies. The study is detailed, precise, and comprehensive.

Introduction is well written. One suggestion, in introduction there is no need to write a full name of genes in capital letters.

Presented results illustrate adequately the gene sequences, as well as associated dental anomalies. Just a suggestion, it is not quite clear from Table 1. as presented, do all patients from family 1 have variant c.722G>A, p.Arg241Gln in KCTD1, and for family 2  c.727C>T, p.Pro243Ser.

Results are discussed thoroughly with adequate conclusion.

Paper needs to be checked for typing errors.

I suggest for this paper to be accepted after minor revision.

Author Response

Response to the comments of the reviewer 1

The authors made quite an effort to conduct genetical study of finding association between genetic variants in KCTD1 and isolated dental anomalies. The study is detailed, precise, and comprehensive.

Thank you very much for your kind words on our work and also for the valuable comments. We have tried our best to correct the manuscript accordingly.

Introduction is well written. One suggestion, in introduction there is no need to write a full name of genes in capital letters.

RESPONSE

Thank you for your comment. We corrected as you suggested.  It is written as follows.

Genetic variants in the Potassium Channel Tetramerization Domain-Containing Protein 1 gene(KCTD1; MIM  613420) are implicated in Scalp-Ear-Nipple or Finlay-Marks syndrome (SEN; MIM 181270), which is characterized by cutis aplasia of the scalp, malformations of breasts, external ears, digits, and nails, as well as renal and dental anomalies [10-14].   Dental anomalies found in patients with Scalp-Ear-Nipple syndrome include microdontia, tooth agenesis, and enamel defects. [10,13,15-18]. To date, thirteen genetic variants in KCTD1, all of which impact the N-terminus of the protein,have been reported to be associated with Scalp-Ear-Nipple syndrome [10-14].

The KCTD1 gene encodes the KCTD1 protein, which directly binds to β-catenin through its BTB (Broad-complex, Tramtrack, and Bric-a-brac) domain, enhances its degradation and inhibits canonical WNT/β-catenin signaling [19,20]. KCTD1 also functions as a transcriptional repressor, which interacts with Transcription factor AP2-alpha (TFAP2A; MIM 107580), Transcription factor AP2-beta (TFAP2B; MIM 601601), and Transcription factor AP2-gamma (TFAP2C; MIM 601602) via its BTB domain and acts as a strong repressor of TFAP2A transcriptional activity [12,19]. Therefore, KCTD1 functions as a potent transcriptional repressor of TFAP2A and a negative regulator of WNT/β-catenin signaling, which is important for embryonic development and homeostatic self‑renewal in tissue regeneration and repair [8,21]. Of note, KCTD1 has also been reported to function as a suppressor of the hedgehog pathway [22].

Presented results illustrate adequately the gene sequences, as well as associated dental anomalies. Just a suggestion, it is not quite clear from Table 1. as presented, do all patients from family 1 have variant c.722G>A, p.Arg241Gln in KCTD1, and for family 2  c.727C>T, p.Pro243Ser.

RESPONSE

Thank you so much for this comment. For clarification, the variant c.722G>A, p.Arg241Gln in KCTD1 is only present in patients 1 (I-2), patient 2 (II-1), patient 3 (II-2), patient 4 (II-3), patient 5 (II-4), and Patient 6 (II-5)family 1 in Table 1. The mother (I-1) and her daughter (II-6) in family 1 do not have this variant. Regarding family 2, the variant c.727C>T, p.Pro243Ser in KCTD1 is present only in patients 7 (I-1), patient 8 (II-2), and patient 9 (II-3). The father, a daughter (II-1) and a son (II-4) in family 2 do not have this variant.

Results are discussed thoroughly with adequate conclusion.

Paper needs to be checked for typing errors.

RESPONSE

Thank you for the valuable comments.

Reviewer 2 Report

Comments and Suggestions for Authors

Dear Authors,

This manuscript explores the genetic variants in KCTD1 and their associations with isolated dental anomalies. This study is interesting and important for future research. My remarks are the following:

Abstract:

--> Nothing to report.

Introduction:

--> Line 68: “KCTD1 encodes KCTD1” – please rephrase, for a better understanting.

Results, Discussion and Conclusions

--> Section 2.1 – a brief description regarding the rest of 353 patients would complete the clinical overview of your entire study lot; as written in the abstract, they also have dental anomalies.

--> Section 2.4 – CADD and DANN complete wording should be present with their first apparition, in line 156 (Section 2.2), not in lines 196-198 (Section 2.4).

--> Line 202: the link should be written as the others, in smaller fonts

--> Caption Figure 6 – be more precise with the statistical results (define the real p value, not just the fact that it is ≤ 0.05, especially since you have a very small sample size); define also the Mann-Whitney U value and the standardized test statistic value. In this case, it would be beneficial to move these details in the text above, not in the figure caption, since in the above paragraph you only mention results that are not statistically significant.

--> You should mention the strong points and weaknesses of your study.

Data and method:

--> Nothing to report.

Best regards!

Comments on the Quality of English Language

Author Response

Response to the comments of the reviewer 2

This manuscript explores the genetic variants in KCTD1 and their associations with isolated dental anomalies. This study is interesting and important for future research.

Thank you very much for the valuable comments. We have tried our best to correct the manuscript accordingly.

My remarks are the following:

Abstract:

Nothing to report.

Introduction:

Line 68: “KCTD1 encodes KCTD1” – please rephrase, for a better understanding.

RESPONSE

Thank you for the comment. We have now written it as “KCTD1 gene encodes KCTD1 protein” to clarify.

Results, Discussion and Conclusions

Section 2.1 – a brief description regarding the rest of 353 patients would complete the clinical overview of your entire study lot; as written in the abstract, they also have dental anomalies.

RESPONSE

Thank you for this valuable comment. It is corrected and written as follows.

4.2. Patient recruitment and clinical investigations

Inclusion criteria for patient recruitment were patients with isolated dental anomalies, including: tooth agenesis, microdontia, macrodontia, supernumerary teeth, odontomas, talon cusps, enamel hypoplasia, tooth fusion, dens evaginatus, failure of tooth eruption, and root anomalies. Oral and radiographic examinations were performed on the cohort of 362 patients who came to The Pediatric Dental Clinic, Faculty of Dentistry, Chiang Mai University (Figure 9). The participants included 196 (54.14%) females and 166 (45.86%) males. The genetic studies of this cohort have been published [6,26,50].

CADD and DANN complete wording should be present with their first apparition, in line 156 (Section 2.2), not in lines 196-198 (Section 2.4).

RESPONSE

Noted with thanks. It is corrected and written as

In a multiple sequence alignment of LRP5 proteins from 494 species of vertebrates, the amino acid residue Gly695 is found in 117 species of vertebrates, suggesting it is not well conserved. The LRP5variant c.2083G>A; p.Gly695Arg is predicted to be a polymorphism (prob = 0.860781214034341) and benign (0.003) by MutationTaster and PolyPhen-2, respectively. The Combined Annotation Dependent Depletion (CADD) and Deleterious Annotation of genetic variants using Neural Networks (DANN) scores for this variant are 21.8 and 0.9059, respectively. Since genetic variants in LRP5 have been reported to be associated with tooth agenesis, taurodontism, and oral exostoses, it is possible that thisLRP5 variant contributes to the dental phenotypes and torus palatinus in patient 7. However, the variant was not found in other affected family members, ruling it out as the primary causal variant in this family.

Line 202: the link should be written as the others, in smaller fonts

RESPONSE

Noted with thanks. It is corrected and written as follows.

2.4. p.Pro243Ser variant

The heterozygous missense variant c.727C>T; p.Pro243Ser was identified in patients 7, 8 and 9, of family 2 who had tooth agenesis, taurodontism, a supernumerary tooth, unseparated roots of permanent molars, and a torus palatinus (Figure 4b). This variant is not reported in gnomAD (https://gnomad.broadinstitute.org; accessed 24 September 2023), LOVD (https://www.lovd.nl; accessed 24 September 2023), HGMD (https://www.hgmd.cf.ac.uk/ac/index.php; accessed 24 September 2023), or the Thai                reference exome variant database(T-REx) [29], therefore it is considered novel. The amino acid residue Pro 243 is highly conserved, being present in 289 species of vertebrates, a selection of which are shown in Figure 5. This variant is predicted to be a polymorphism by MutationTaster (0.931756239937238) (https://www.mutationtaster.org; accessed on 24 September 2023, benign by PolyPhen-2 (0.007) (http://genetics.bwh.harvard.edu/pph2/; accessed on 24 September 2023), and tolerated by Sorting Intolerant From Tolerant (0.1) (SIFT; https://bio.tools/sift; accessed on 24 September 2023). The CADD (https://cadd.gs.washington.edu; accessed on 24 September 2023) and DANN (https://cbcl.ics.uci.edu/public_data/DANN/; accessed on 24 September 2023) scores for this variant are 16.16 and 0.9425, respectively, suggesting it may be benign (https://varsome.com/about/resources/germline-implementation/). Neither variant has any                   predicted cryptic impact on KCTD1 mRNA splicing as determined using SpliceAI or Pangolin (https://spliceailookup.broadinstitute.org/).

Caption Figure 6 – be more precise with the statistical results (define the real p value, not just the fact that it is ≤ 0.05, especially since you have a very small sample size); define also the Mann-Whitney U value and the standardized test statistic value. In this case, it would be beneficial to move these details in the text above, not in the figure caption, since in the above paragraph you only mention results that are not statistically significant.

RESPONSE

We thank the reviewer for the comments. The actual P-value was indicated in a revised Figure 6b. The figure legend was updated as follows. Please see on page 8, line 220-223 in the revised manuscript.

Figure 6. Immunoblotting of KCTD1. (a) Image of typical immunoblot bands of wild-type KCTD- WT, p.Arg241Gln, and p.Pro243Ser variants, along with β-catenin and β-tubulin. (b) The fold change in the intensity of the bands relative to the wild-type KCTD1 was compared and shown in the bar chart (means and standard deviations of four experiments). Statistical significance is indicated by *.

Additionally, the Mann-Whitney U value and the standardized test statistic Z value were defined in statistical analysis method under the Materials and Methods section of the revised manuscript. Please see on page 15, lines 465-468 in the revised manuscript.

It is written as follows.

Statistical analysis

Data are presented as means ± standard deviations (SD). GraphPad Prism 9.4.1 was used for statistical analysis of band intensities (n=4) using the Mann-Whitney U test to compare differences between two independent groups. A P-value ≤ 0.05 was considered to be statistically significant.

You should mention the strong points and weaknesses of your study.

RESPONSE

Thank you for this suggestion. The strong points and the limitation of the study are written as follows (Lines 478-507).

  1. Limitations of the study

     Cone beam computed tomography would have been preferable to further characterize the root maldevelopment seen in all patients with the KCTD1 variant, but this technology was not clinically available at the times of assessment. Functional studies of p.Pro243Ser did not show significant effects on β-catenin levels. However, our SUMOplot analysis identified a predicted consensus sequence for SUMOylation, ψKXE (with sumoylation at K252), in the C-terminal tail of KCTD1. Therefore, functional studies to test the effect of the p.Pro243Ser variant on SUMOylation may have supported the effects of this variant on tooth development. Lastly, functional studies of the effects of both KCTD1 variants on SHH signaling could be performed in the future to directly test the effects of these variants on this pathway that also plays a critical role in tooth development. 

  1. Conclusion

In summary, we present evidence to support the contribution of KCTD1 variants to the presentation of isolated dental anomalies and oral exostoses in our two families because 1) both variants are rare (p.Arg241Gln) or novel (p.Pro243Ser); 2) the variants fully segregated with the phenotypes in each family; 3) previously reported patients with KCTD1 mutations-associated Scalp-Ear-Nipple syndrome were reported to have dental anomalies, indicating the link between KCTD1 and tooth development; 4) the location of our identified variants is in the C-terminal end of KCTD1, compared to those of patients with Scalp-Ear-Nipple syndrome, who had variants located in the N-terminal BTB domain; and 5) no other rare variants in the known dental anomalies-associated genes were identified to cosegregate with dental anomalies in the families. This study demonstrates for the first time that genetic variants in KCTD1 are associated with isolated dental anomalies. This is supported by the functional studies which showed that the p.Arg241Gln KCTD1 variant affected β-catenin levels, an effector of Wnt/β-catenin signaling. Both variants (p.Arg241Gln and p.Pro243Ser) are located in the C-terminal tail of the protein, suggesting a possible genotype-phenotype correlation. Further studies addressing the impact of the variants on protein-protein interactions, protein stability, and SUMOylation, and subsequent aberrant WNT-SHH-BMP signaling are needed.

Data and method:

Nothing to report.
